# Effect of the Addition of GGBS on the Frost Scaling and Chloride Migration Resistance of Concrete

**Vera Correia [1], João Gomes Ferreira [1,\*], Luping Tang [2] and Anders Lindvall [3]**

[1] CERIS, Instituto Superior Técnico, Universidade de Lisboa, 1049-001 Lisboa, Portugal; veramfcorreia@gmail.com

[2] Division of Building Technology, Chalmers University of Technology, 412 96 Göteborg, Sweden; tang.luping@chalmers.se

[3] Thomas Concrete Group, 412 96 Göteborg, Sweden; anders.lindvall@c-lab.se

\* Correspondence: joao.gomes.ferreira@tecnico.ulisboa.pt; Tel.: +351-218418213

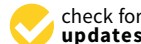

**Featured Application: Use of furnace slag for protecting concrete against deterioration caused by frost scaling cycles and against corrosion of steel rebars caused by chloride attack.**

**Abstract:** Ground Granulated Blast-furnace Slag (GGBS) can partially replace cement in concrete to improve certain properties. However, some concerns regarding its performance have been raised. This research aimed at investigating the properties of concrete with GGBS, with special focus on its frost scaling and chloride ingress resistance. Concretes with different amounts of GGBS, different efficiency factors, and different air contents have been tested. The effects of other factors, namely the curing temperature, the use of superplasticizer and carbonation, have also been investigated. The results showed that the frost resistance generally decreases with the increase of the amount of GGBS. However, this research showed that it is possible to produce frost resistant concrete with up to 50% of GGBS by changing some properties of the mix (such as increasing the air content). The results also showed a significant improvement of the chloride ingress resistance for concrete with high additions of GGBS.

**Keywords:** concrete; GGBS; salt frost resistance; chloride migration

## 1. Introduction

The present investigation focused on the behaviour of concrete with addition of Ground Granulated Blast-furnace Slag (GGBS). GGBS, or slag, can be present in concrete as a separate material added in the mixer, or blended in the Portland cement: either as CEM II/A-S or CEM II/B-S (with up to 35% of GGBS of the total binder content) or as CEM III (also called Blast Furnace Slag cement, containing 36% to 95% of GGBS of the total binder content) [1].

GGBS is often used in concrete to improve certain properties, both in fresh and hardened states. GGBS is frequently used in the production of low heat cement. Ternary systems composed by Portland cement, slag and silica fume have also been used in the production of high-strength concrete. However, the main application of GGBS as a supplementary cementitious material is in concrete structures placed in marine environments, due to its proven influence in reducing chloride ingress in concrete. Other beneficial effects of GGBS in concrete consist of protection against sulphate attack and alkali-silica reaction, also widely reported [2].

However, some concerns have been raised regarding the durability of concrete with addition of GGBS in freezing environments, especially when de-icing salts are used. Although in the past years there is a new development by integrating anhydrous sodium acetate into concrete for protection against harmful impact of de-icing salt [3], for conventional concrete structures, the use of de-icing

salts may result in a type of frost attack in concrete called surface scaling. In this type of frost attack, small chips of the cement paste are removed from the concrete surface, exposing the aggregate. The continuation of this mechanism may lead to severe consequences in the structures as the thickness of the protective layer above the reinforcement is reduced, opening an easier path for chloride to reach the steel reinforcement, promoting the initiation of corrosion and consequent loss of load capacity [4].

Several investigations report reduced salt-frost resistance of concrete with additions of GGBS, when compared to that of Portland cement concrete, especially when large additions of GGBS are used. However, this conclusion is not consensual; even though laboratory tests usually result in a poorer performance, many researchers have found that, in field exposure, the resistance of concrete with additions of slag is comparable to that of Portland cement concrete [2].

The research reported herein aimed at investigating the properties of concrete containing additions of GGBS, especially regarding its salt-frost resistance. The use of GGBS in concrete in Sweden is regulated by the Swedish Standard SS 13 70 03 [5], which is the Swedish adaptation of EN 206-1. In this standard, the addition of GGBS is limited to 50 weight-% of CEM I for exposure classes XF1-3, and 25 weight-% of CEM I in exposure class XF4. One of purposes of this study was to investigate whether it is possible to produce GGBS concrete with adequate salt-frost scaling resistance using higher amounts of GGBS than those allowed by SS 13 70 03 [5].

Therefore, air-entrained concrete mixes with an equivalent water/cement ratio of 0.45 and with different percentages of replacement of Portland cement by GGBS were produced and tested. Besides the salt-frost scaling resistance, the compressive strength and resistance against chloride migration were also studied. The influence of the air entrainment and curing temperature on the properties of the hardened GGBS concrete has also been investigated.

## 2. Laboratory Study

### 2.1. Mix Design

The aim of the laboratory study carried out during this research was to evaluate the effect of the replacement of Portland cement by GGBS on the properties of fresh and hardened concrete, with emphasis on its salt-frost scaling resistance and chloride ingress. Air entrained concrete mixes were produced and tested with different amounts of replacement, namely 0%, 25%, 50% and 100% of GGBS by weight of CEM I. It should be noticed that, if not otherwise stated, the addition of GGBS in this paper is referred to the percentage of the cement (CEM I) weight, in accordance with [5].

As the main binder commonly used in Sweden for infrastructural concrete structures for reducing the risks of thermal cracking, alkali-silica reaction and sulphate attacks, a moderate heat, low-alkali, sulphate-resistant Portland cement was used: *Cementa Degerhamn Anläggningscement* (CEM I 42.5 N MH/SR/LA), produced by *Cementa* AB. The coarse aggregates used were Swedish natural and crushed stone, *Tagene* (4–8 mm and 8–16 mm). The fine aggregates used were *Sjösand* (sea sand, 0–4 mm) and *Hol* (0–8 mm).

The Ground Granulated Blast Furnace Slag used was *Slagg Bremen*, imported from *Holcim Deutschland A.G.* The GGBS complies with all the specifications required by EN 15167-1 [6], and was added separately in the mixer.

Four mixes with different GGBS contents were produced, namely:

- Mix 1: 0% GGBS, reference;
- Mix 2: 25% GGBS (which is the maximum amount of GGBS allowed for XF4 exposure class according to SS 13 70 03 [5]). This corresponds to an addition of 20% GGBS of the binder weight (i.e., a CEM II/A-S);
- Mix 3: 50% GGBS. This corresponds to an addition of 35% GGBS of the binder weight (i.e., a CEM II/B-S);
- Mix 4: 100% GGBS. This corresponds to an addition of 50% GGBS of the binder weight (i.e., a CEM III/A).

The equivalent water/cement ratio used was 0.45 for all the mixes, which is the maximum allowed in SS 13 70 03 [5] for exposure class XF4. In the calculation of the equivalent water/cement ratio for mixes with GGBS, an efficiency factor of 0.6 was considered, which is the highest k-factor allowed for GGBS added in the mixer together with CEM I, according to SS 13 70 03 [5].

This k-factor, or efficiency factor, is a measure of the relative contribution of the addition to the strength of concrete, compared to an equivalent mass of Portland cement.

The efficiency factor concept is based on the fact that, since the hydration of the GGBS is slower than the hydration of Portland cement, if CEM I is replaced by slag on a one-to-one basis, the strength at early ages will be lower. Since the compressive strength of concrete is highly dependent on the water/cement ratio, it follows that one way to achieve higher strength at early ages when using GGBS is to decrease the ratio of water/(cementitious material) content, i.e., the mass of GGBS used will be higher than the mass of Portland cement that it is replacing [7].

This means that $x$ kg/m$^3$ of GGBS is equivalent to $kx$ kg/m$^3$ of Portland cement, to achieve the same compressive strength as Portland cement concrete at a given age (usually 28 days), and the total equivalent cement content is C + $k$S, being C the mass of cement and S the amount of slag. Therefore, instead of using the water/cement ratio or water/binder ratio, the equivalent water/cement ratio ($(w/c)_{eq}$) is used, which is given by Equation (1):

$$(w/c)_{eq} \ = \ \frac{w}{c + 0.6\,S} \tag{1}$$

where "$w$", "$c$" and "$S$" are, respectively, the weight of water, cement and slag per unit volume.

All concrete mixes were air entrained. The air content was targeted at 4.5% ± 0.5%. The targeted air content for each mix was achieved using a synthetic surfactant type of air entraining agent by Sika, SikaAer-S (1:10). SikaAer-S is produced by Sika Sverige AB in Sweden.

In order to evaluate the effect of an increased air content on the different properties of the hardened concrete, particularly the salt-frost scaling resistance of concrete with GGBS, an additional mix with 50% of slag replacement and targeted air content of 6.0% ± 0.5% was produced (Mix 5).

Other additional concrete mix (Mix 6) with 50% GGBS was defined, by using a k-factor of 1, i.e., in this mix, the mass of slag replaces the exact same reduced mass of Portland cement.

The proportion of each concrete mix was chosen so that its consistency would correspond to an S3 class of slump (slump between 100 and 150mm). The desired slump was achieved by using a superplasticizer, namely Sikament 56/50, a third generation Polycarboxylate Ether (PCE) based superplasticizer produced by Sika Sverige AB.

Summarizing, a total of six different mixes were produced:

- Mix 1: 0% GGBS, 4.5% ± 0.5% Air (reference concrete);
- Mix 2: 25% GGBS, 4.5% ± 0.5% Air, k = 0.6;
- Mix 3: 50% GGBS, 4.5% ± 0.5% Air, k = 0.6;
- Mix 4: 100% GGBS, 4.5% ± 0.5% Air, k = 0.6;
- Mix 5: 50% GGBS, 6.0% ± 0.5% Air, k = 0.6;
- Mix 6: 50% GGBS, 4.5% ± 0.5% Air, k = 1.0.

The proportions of all the components of the final mixes are presented in the Table 1 below:

**Table 1.** Final mix design.

| | Mix | | | | | |
|---|---|---|---|---|---|---|
| | 1 | 2 | 3 | 4 | 5 | 6 |
| Amount of GGBS (% of CEM I) | 0 | 25 | 50 | 100 | 50 | 50 |
| k-factor | | 0.6 | 0.6 | 0.6 | 0.6 | 1 |
| Targeted air content [+/-0,5%] (%) | 4.5 | 4.5 | 4.5 | 4.5 | 6 | 4.5 |
| $(w/c)_{eff}$ ratio | 0.45 | 0.52 | 0.59 | 0.72 | 0.59 | 0.68 |
| $(w/b)$ ratio | 0.45 | 0.41 | 0.39 | 0.36 | 0.39 | 0.45 |
| $(w/c)_{eq}$ ratio | 0.45 | 0.45 | 0.45 | 0.45 | 0.45 | 0.45 |
| Cement [kg/m$^3$] | 390 | 330 | 330 | 280 | 330 | 250 |
| GGBS [kg/m$^3$] | 0 | 82.5 | 165 | 280 | 165 | 125 |
| (GGBS/total binder) ratio | 0 | 0.2 | 0.33 | 0.5 | 0.33 | 0.33 |
| Equivalent cement content | 390 | 379.5 | 429 | 448 | 429 | 375 |
| Aggregate | | | | | | |
| 　Sjösand (0–4 mm) [kg/m$^3$] | 449.8 | 446.4 | 411 | 388.6 | 410.7 | 455 |
| 　Hol (0–8 mm) [kg/m$^3$] | 274.7 | 272.7 | 251 | 237.3 | 250.8 | 277.9 |
| 　Tagene (4–8 mm) [kg/m$^3$] | 121.6 | 120.7 | 111.1 | 105 | 111 | 123 |
| 　Tagene (8–16 mm) [kg/m$^3$] | 885.7 | 879 | 809.2 | 765.1 | 808.7 | 896 |
| Water [kg/m$^3$] | 175.5 | 170.8 | 193.1 | 201.6 | 193.1 | 168.8 |
| AEA [kg/m$^3$] | 0.975 | 0.99 | 1.155 | 1.12 | 2.475 | 1 |
| AEA [% of cement by weight] | 0.25 | 0.3 | 0.45 | 0.4 | 0.75 | 0.4 |
| AEA [% of binder by weight] | 0.25 | 0.3 | 0.35 | 0.4 | 0.75 | 0.4 |
| Plasticizer [kg/m$^3$] | 1.365 | 1.32 | 2.145 | 1.4 | 1.155 | 1.125 |
| Plasticizer [% of cement by weight] | 0.35 | 0.4 | 0.65 | 0.5 | 0.35 | 0.45 |

For each concrete quality, two mixes of about 30 litres were produced using a Zyklos rotating pan mixer. The surface of the mixer was moist, and all dry materials (aggregates and binders) were added, followed by the water. After mixing these components for 30 s, the air entraining agent and the superplasticizer were added, and the concrete was mixed for another 120 s. Directly after mixing, the following tests on the freshly mixed concrete were performed:

- Slump test, according to SS-EN 12350-2 (Figure 1);
- Air content, according to SS-EN 12350-7 (Figure 2);
- Air void analysis (AVA) in the fresh state (Figure 3).

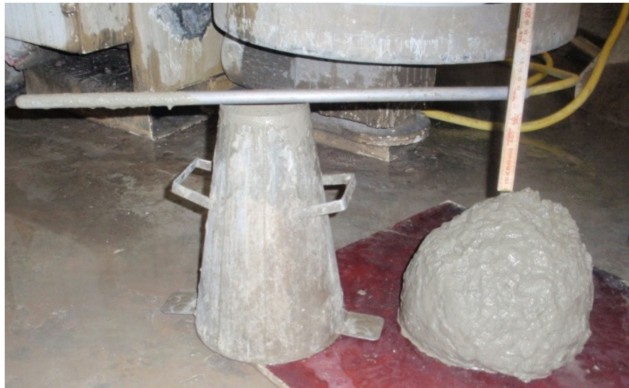

**Figure 1.** Slump test apparatus.

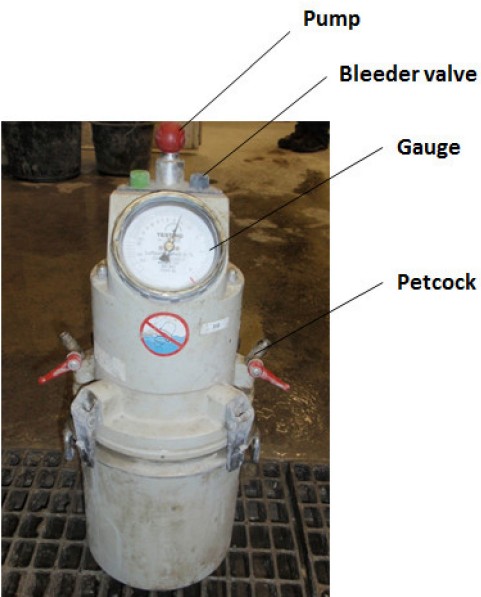

**Figure 2.** Vessel used for measuring the total air content of concrete in the fresh state, according to standard SS-EN 12350-7 [8].

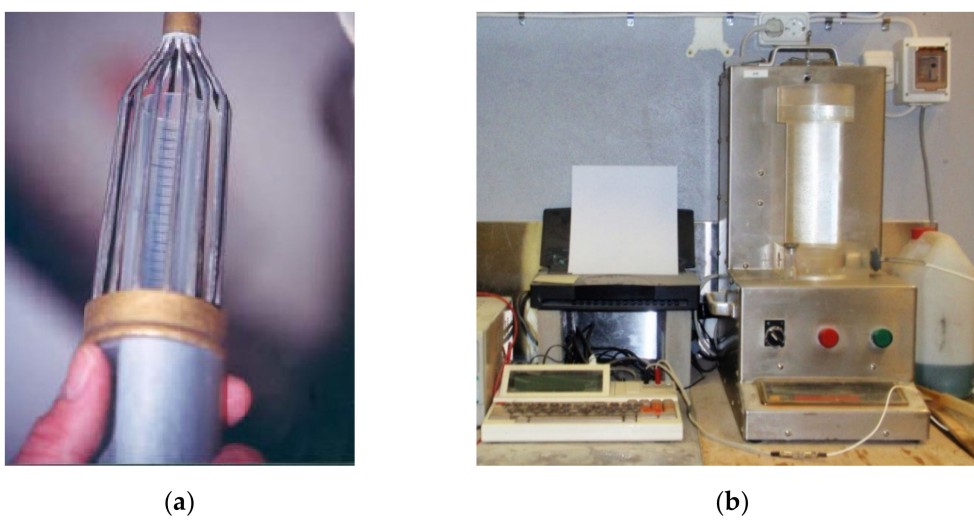

(**a**)                                                                                      (**b**)

**Figure 3.** Apparatus for air void analysis (AVA) in the fresh state: (**a**) Wire cage surrounding the syringe used to collect the sample for the AVA test; (**b**) AVA test set-up. Riser column with inverted pan on (on the right) and computer and printer that process the information (on the left).

After mixing, concrete was cast in cylindrical and cubic moulds that comply with specifications in SS-EN 12390-1 for testing hardened concrete. After 24 h, the specimens were removed from the moulds and curing (in accordance with the specifications of different test methods) began. After curing was completed, the hardened concrete specimens were subjected to the following tests:

- Compressive strength, according to SS-EN 12390-3;
- Rapid chloride migration test, according to NT Build 492;
- Salt-frost scaling, according to SS 13 72 44 [9]. This standard corresponds to CEN/TS 12390-9:2016.

*2.2. Rapid Chloride Migration According to NT Build 492*

The resistance against chloride ingress was determined by the non-steady state migration test described in NT Build 492, which determines the chloride migration coefficient of hardened concrete

specimens from non-steady-state migration experiments. In this method an external electrical field is applied across a concrete specimen of size ∅100 × 50 mm, forcing the chloride ions to migrate into the specimen. After a certain period of time, the specimen is split and the chloride penetration depth is measured by spraying a silver nitrate solution on the split sections. The non-steady state migration coefficient is then calculated based on the penetration depth.

Figures 4 and 5 show, respectively, the test set-up and the tested specimens, after being sprayed with the silver nitrate solution.

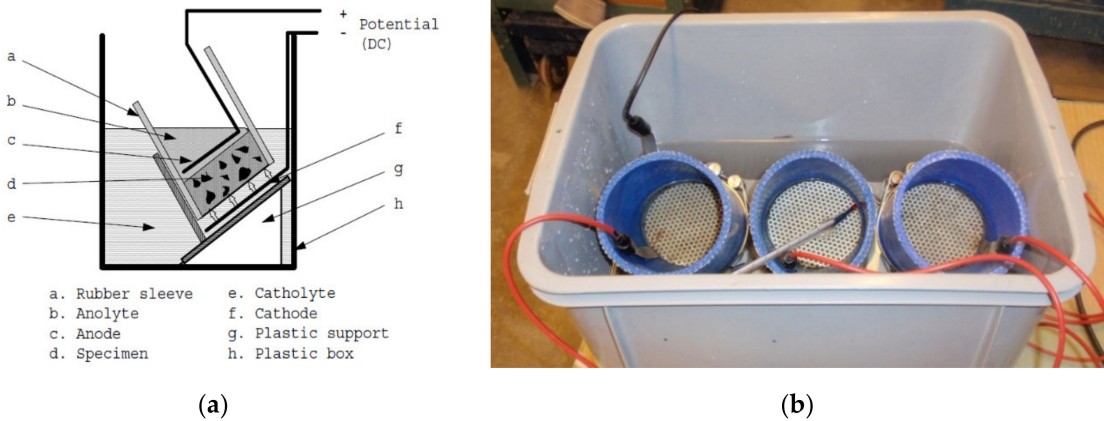

(**a**)　　　　　　　　　　　　　　　　　　　　　　　　(**b**)

**Figure 4.** Test setup of the Rapid Chloride Migration test. (**a**) Schematic representation [10]; (**b**) Execution of the tests.

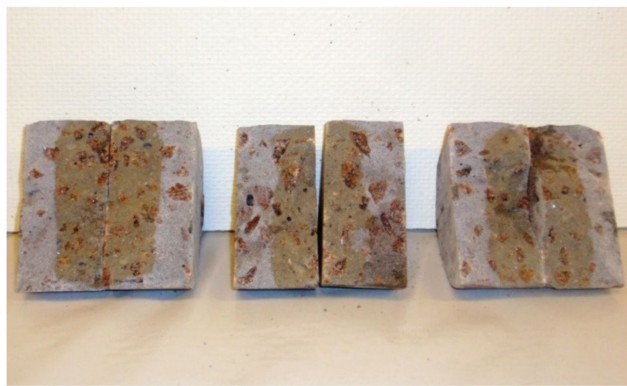

**Figure 5.** Test specimens for the Rapid Chloride Migration test after being sprayed with silver nitrate solution.

*2.3. Scaling under Freezing and Thawing According to SS 13 72 44*

The Swedish Standard SS 13 72 44 (2005) is used to determine the resistance to scaling of a horizontal concrete surface exposed to freezing and thawing cycles with or without the presence of de-icing chemicals.

Concrete cubes 150 × 150 × 150 mm are cured in water 20 ± 2 °C until the 7th day of age, after which they are placed in a climate chamber at 20 ± 2 °C and 65% ± 5% R.H. At 21 ± 2 days of age, a 50 mm slab is sawn of the cube. The specimen is then returned to the climate chamber for 7 days, during which a rubber sheet is glued in all surfaces of the specimen except the sawn surface as test surface.

At 28 days of age, the specimen is re-saturated by pouring tap water onto the test surface. After 72 ± 2 h, the tap water is removed and the freezing medium (saline solution with 3% NaCl) is applied at a depth of 3 mm. The specimen is thermally insulated with 20 mm of polystyrene, and covered with a tight plastic foil to prevent evaporation. The test set-up is shown in Figure 6. The specimen is then placed in the freezer and the test begins, at the 31st day of age.

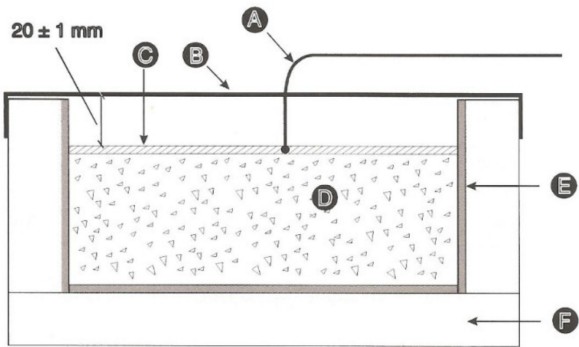

**Figure 6.** Freeze/thaw test set-up. A: Thermo element; B: Protection against evaporation; C: Freezing medium; D: Test specimen; E: Rubber cloth; F: Thermal insulation. [9].

In the freezer, the specimens are subjected to repeated freezing and thawing. The temperature in the freezing medium cycles from -20 °C to 20 °C over a period of 24 h, with the temperature exceeding 0 °C for at least 7 h, but not more than 9 h in each cycle.

After 7, 14, 28, 42, 56, 70, 84, 98 and 112 cycles, the scaled-off material is collected and weighted. The results are expressed as accumulated mass of scaled material per area of test surface as function of freeze/thaw cycles. At every 7th cycle, the scaled material is collected and weighted, and the mass loss per area, $m_n$ (kg/m$^2$), is determined and registered, according to Equation (2),

$$m_n = \frac{M_n}{A} \tag{2}$$

where $m_n$ is the accumulated mass of scaled material (kg) after n cycles and $A$ is the area of the test surface (m$^2$).

Figure 7 shows the tests preparation phase, while Figure 8 presents the weighting of the scaled material.

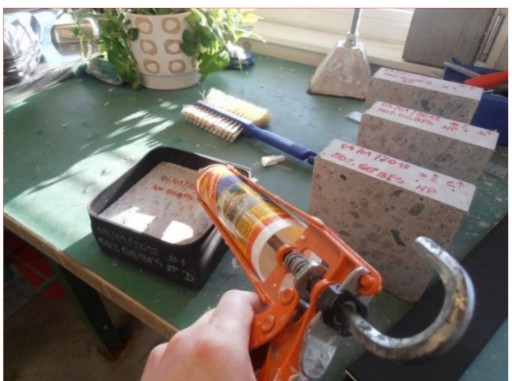
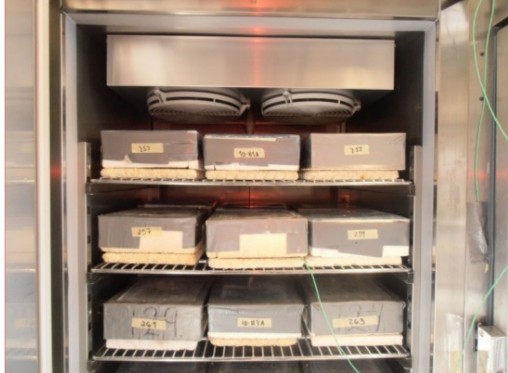

**Figure 7.** Preparation of salt-frost scaling tests. **Left**: Application of the rubber cloth and silicone sealant on the test specimens; **Right**: Salt-frost scaling test specimens placed in the climate chamber.

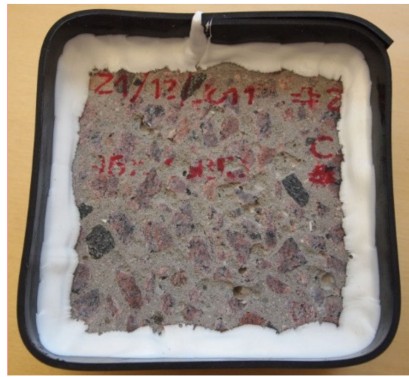 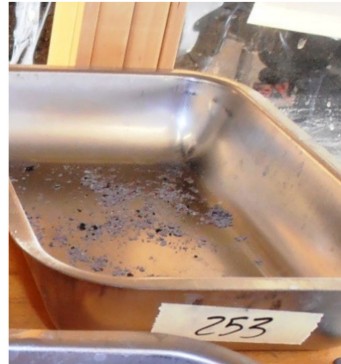 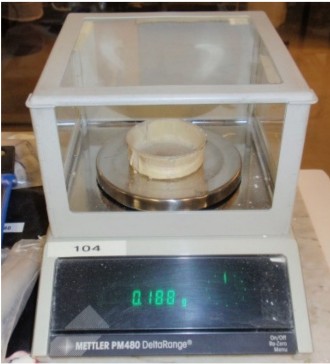

**Figure 8. Left**: Example of specimen after 112 freeze/thaw cycles (specimen of Mix 2%-25% GGBS, 4.5% Air, k = 0.6); **Center**: Scaled-off material collected on a steel vessel; **Right**: Determination of the mass of scaled material.

The frost resistance of a concrete quality is evaluated according to the following criteria (Table 2):

**Table 2.** Acceptance criteria for the frost scaling resistance of concrete according to SS 13 72 44 (2005).

| Frost Resistance | Requirements |
|---|---|
| Very good | The mean value of the scaled material after 56 cycles ($m_{56}$) is less than 0.10 kg/m$^2$. |
| Good | The mean value of the scaled material after 56 cycles ($m_{56}$) is less than 0.20 kg/m$^2$ **and** $m_{56}/m_{28}$ is less than 2; or The mean value of the scaled material after 112 cycles ($m_{112}$) is less than 0.50 kg/m$^2$. |
| Acceptable | The mean value of the scaled material after 56 cycles ($m_{56}$) is less than 1.00 kg/m$^2$ **and** $m_{56}/m_{28}$ is less than 2 Or The mean value of the scaled material after 112 cycles ($m_{112}$) is less than 1.00 kg/m$^2$. |
| Unacceptable | The requirements for acceptable frost resistance are not met. |

The number of specimens per test, the age at which they were tested and the geometry of the specimen are described in Table 3.

**Table 3.** Specimens casted, geometry of the moulds and age of testing of the hardened concrete.

| Test In The Hardened Concrete | Age (Days) | Number of Specimens | Geometry (mm) |
|---|---|---|---|
| Compressive Strength (EN 12390-2) | 7 | 2 | |
| | 28 | 3 | |
| | 56 | 2 | |
| Rapid Chloride Migration (NT Build 492) | 28 | 1 | Cylinder, 100 × 50 |
| | 56 | 1 | (Ø x h) |
| Scaling under freeze/thaw (SS 13 72 44) | 31 | 4 | 150 × 150 × 50 slabs cut from cube, 150 (e) |

GGBS concrete presents a slower rate of hydration than Portland cement concrete, which may lead to a lower degree of hydration at the age of 28 days. A lower degree of hydration usually results in a more porous concrete, with lower compressive and tensile strengths. Given that the freeze/thaw test starts at 31 days of age, the salt-frost resistance of concrete with GGBS may be adversely affected by its lower hydration degree, when compared with Portland cement concrete. In order to evaluate the effect of a more prolonged hydration of the concrete specimens with GGBS before being exposed to the freeze/thaw cycles, 3 specimens from each of the mixes with 50% of GGBS replacement and different air content (Mix 3 and Mix 5) were subjected to prolonged pre-treatment. From each cube of these mixes, two slabs (instead of one) were cut and tested. One slab was pre-conditioned according to the standard procedure described in SS 13 72 44. The other slab was kept in the climate chamber for

14 more days, and only afterwards the water was poured onto the test surface. For these specimens, the test started at the 45th day of age, instead of the 31st day.

## 3. Results and Discussion

The main results obtained during the present investigation are presented in this section. Only the results of the hardened state will be displayed. Reference to the results of the tests performed in the fresh concrete will be made when the influence of any of the parameters measured is of interest to discuss the results obtained for the hardened concrete specimens.

### 3.1. Compressive Strength

The results obtained for the compressive strength of the concrete mixes, determined according to SS-EN 12390-3 (Figure 9) are presented below.

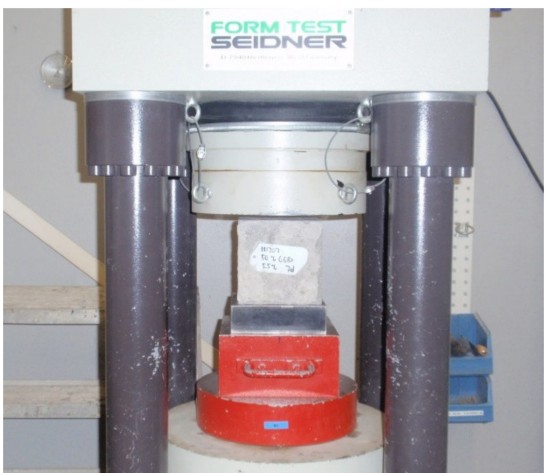

**Figure 9.** Compressive strength test set-up.

### 3.1.1. Influence of the Amount of Portland Cement Replacement by GGBS

As shown in Figure 10, the compressive strength at 7 days of age is higher for Portland cement-only concrete, and decreases with an increase in the addition of GGBS. The rate of hydration of GGBS is usually lower than that of Portland cement, which results in a lower rate of strength gain for concrete with additions of slag. Consequently, GGBS concrete will present a lower compressive strength at early ages. Furthermore, given that the reactivity of the slag is lower than that of the Portland cement concrete, the compressive strength at early ages will be lower for concretes in which the contribution of slag for the compressive strength is higher, i.e., concrete mixes where slag replaces a higher amount of cement [2].

Even though the rate of strength development is different for all the mixes, the average compressive strength obtained for all concrete qualities at 28 days is very similar. In the present investigation, the k-factor concept was used. The efficiency factor is used in order to obtain similar compressive strength at 28 days for concrete with additions as the comparable Portland-cement concrete. The results obtained show the adequacy of using the efficiency factor in the production of concrete with addition to obtain similar compressive strength at 28 days of age. The efficiency factor is, therefore, used to offset the slower hydration of the supplementary cementitious materials, which results in a higher compressive strength than it would be obtained if slag would replace Portland cement on a one-to-one basis.

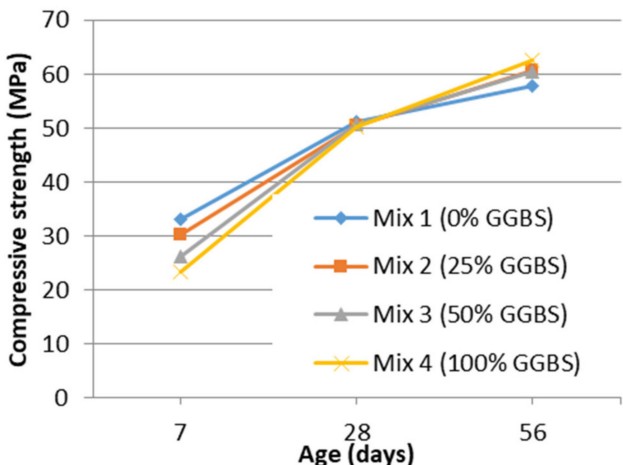

**Figure 10.** Compressive strength development for concretes with different addition percentages of GGBS, k = 0.6, targeted air content of 4.5% ± 0.5%, cured at 20 °C.

The results also show that, for 56 days of age, the compressive strength increases with an increase in slag content, at least for concrete with percentages of replacement of 100% by mass of GGBS of the Portland cement content (50% of the total binder content), with the reference Portland cement concrete presenting the lowest compressive strength. These results are also in agreement with the literature. The hydration of GGBS continues for longer periods of time, when compared to the hydration of Portland cement, resulting in a higher compressive strength of the GGBS concrete at later ages. On the other hand, the denser microstructure of GGBS concrete also contributes to a higher strength, as long as sufficient hydration has occurred [4].

### 3.1.2. Influence of the Air Content of Concrete

Figure 11 presents the results for the compressive strength at 7, 28 and 56 days of age, for concrete mixes with 50% GGBS produced with different targeted air contents (4.5% ± 0.5% for Mix 3, and 6.0% ± 0.5% for Mix 5).

The results show that the concrete mix with higher amount of air (Mix 5, with air content between 5.5% and 5.6%, measured according to SS-EN 12350-7) presents lower compressive strength at all ages when compared to Mix 3 (air content measured between 4.1% and 4.9%). Since the compressive strength of concrete is directly related to its density and an increase in the air voids will reduced the density of the paste, an increase in the air content is expected to lead to a lower compressive strength [4].

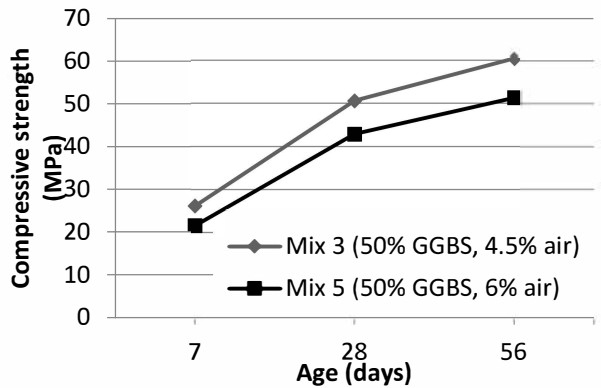

**Figure 11.** Compressive strength development of concretes with 50% GGBS, k = 0.6, cured at 20 °C, and with different air contents.

According to the literature, a 1% increase in the volume of air pores in concrete results in an average compressive strength loss between 5.5% [4] and 6.0% [7]. Considering the average value of 4.5% for the air content of Mix 3, and 5.6% for Mix 5 (results obtained with the pressure gauge method described in SS-EN 12350-7), a difference of 1.1% is obtained. For this difference, however, the results obtained in compressive strength tests show a drop of between 15% and 17%, for all ages of testing. This value is much higher than the 6.0% mentioned in the literature. The reasons for these results are unclear, but might be related with the accuracy of the measurements of the air content of the concrete qualities. In fact, the results obtained with the AVA show a higher air content for both mixes (in comparison with the targeted values), with Mix 3 presenting an average air content of 5.9% and Mix 5 and average air content of 9.3%. Considering an average decrease of 5.5% in the compressive strength for each 1% of increase in the air content, a total decrease of approximately 19% is obtained for Mix 5, which is closer to the results obtained. These results show that there is a possibility that the results for the air content measured with the AVA are more accurate than the results obtained with the pressure gauge method, at least for these mixes. An air void analysis of the hardened concrete could have been carried out to assess the actual air content.

### 3.1.3. Influence of Efficiency Factor

The results displayed in Figure 12 show that Mix 6, with a k-factor of 1, presents a lower compressive strength at all ages of testing when compared with Mix 3, produced with an efficiency factor of 0.6.

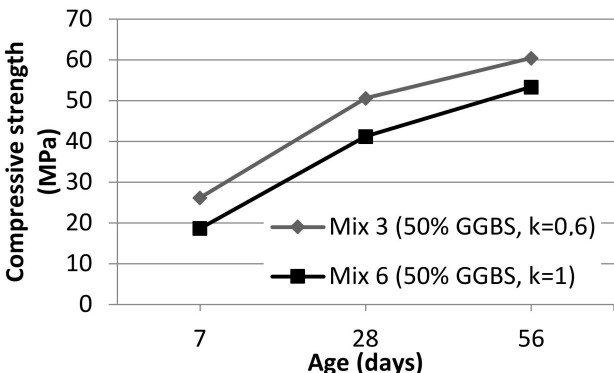

**Figure 12.** Compressive strength development of concretes with 50% GGBS, air = 4.5%, cured at 20 °C and with different efficiency factors.

The compressive strength of concrete with additions depends on the activity index of the addition. Given that the rate of strength development of concrete with GGBS is lower than that of Portland cement-only concrete, an efficiency factor concept was developed. In addition, the compressive strength of concrete highly depends on the water/cement ratio. Therefore, one way to achieve the same strength when using additions is to decrease the water/binder ratio. This means that, in order to achieve similar compressive strength at 28 days, for the same water content, the mass of addition used must be larger than the mass of cement that it is replacing. Thus, according to the k-factor concept, the mass of addition (A) replaces a mass of kA of Portland cement. In this approach, instead of using the water/binder ratio, an equivalent water/cement ratio is used, which is defined by Equation (1) [7]. A k-factor of 1 means that the addition replaces the exact same amount of cement, which is the case for Mix 6. A k-factor lower than 1 means a larger amount of binder material and, therefore, a lower effective water/binder ratio (Mix 3). In this research, a k-value of 0.6 was used, as prescribed in [5] for GGBS additions. Therefore, even though the equivalent water/cement ratio is 0.45 for both mixes, for Mix 3 (with an efficiency factor of 0.6), the water/binder ratio is 0.39, whereas for Mix 6 (with a k-factor of 1) the water/binder ratio is 0.45. Given that the compressive strength decreases with an increase

in water/binder ratio [4], Mix 6 would theoretically show lower compressive strength, which is in accordance with the results obtained.

The results presented show, once again, effectiveness of the use of the efficiency factor concept in reducing the influence of the addition of GGBS in the compressive strength of concrete at all ages.

### 3.2. Rapid Chloride Migration

The resistance against chloride ingress was determined by the non-steady state migration test described in NT Build 492.

The results obtained are presented in Figures 13–15, which illustrate the results obtained for the non-steady state migration coefficient, determined according to the above mentioned standard.

### 3.2.1. Influence of the Amount of Portland Cement Replacement by GGBS

The results plotted in Figure 13 show a lower chloride migration coefficient for the specimens tested at 56 days of age than for those tested at 28 days of age, for all concrete qualities. The resistance against chloride ingress depends on the diffusivity/penetrability of concrete, which in turn depends on the degree of hydration of concrete. The flow of liquids (and gases) through the concrete paste is facilitated through the larger capillary pores than through the smaller gel pores. With progress of hydration, the capillary pores in concrete are gradually filled with the C-S-H gel that is formed during the hydration reactions, i.e., the pore structure changes continuously, and the diffusivity of the paste is reduced with progress of hydration. Since the degree of hydration (amount of cement that has reacted, relative to the total amount of cement present in the mix) increases with time, the diffusivity will thus be reduced and, consequently, also the chloride migration coefficient.

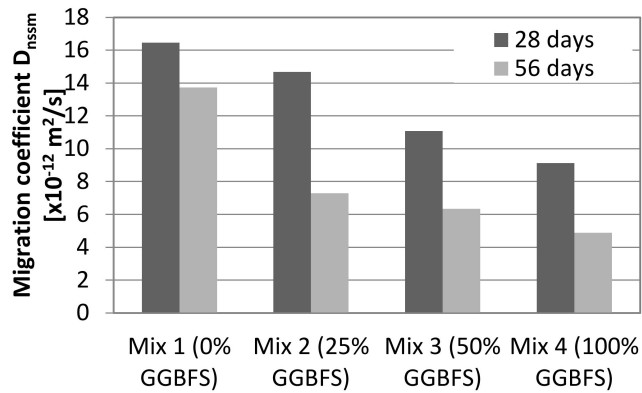

**Figure 13.** Rapid chloride migration coefficient, $D_{RCM}$, for mixes cured at 20°C with different additions of GGBS.

However, the effect of the hydration degree of concrete between 28 and 56 days is more obvious for concrete mixes with GGBS addition than for the reference Portland-cement only concrete. For Portland cement concrete (Mix 1), the DRCM coefficient decreases only 17% between specimens tested at 28 and 56 days of age, whereas for all concrete mixes with GGBS (Mixes 2, 3 and 4), a decrease between 43% and 50% was obtained. These results are related with the lower hydration rate of GGBS, which leads to a larger average pore size at early ages [7]. However, unlike the hydration of Portland cement, whose rate is greatly reduced after 28 days of age, the hydration of GGBS continues at a high rate, which leads to the continuous formation of gel and consequent reduction of the diffusivity of concrete with time [11].

The results also display a clear decrease in the average chloride migration coefficient with the increase in the amount of GGBS replacement in the mix. That is the case for concrete specimens tested at both 28 days and 56 days of age. The positive influence of the increase in GGBS content in the improvement of the resistance against chloride migration is widely reported, and is usually attributed

to the following factors: on the one hand, an increase in the slag content results in a denser and less permeable concrete, which reduces the diffusion of chloride ions and slows down capillary suction, therefore reducing the transport of the chloride ions inside the concrete, and lowering the concentration of available chloride ions in the paste [4,12–14]. On the other hand, the lower concentration of hydroxyl ions in the pore solution of the GGBS concrete reduces its capacity to exchange anions, which also contributes for the reduction of the diffusion rate of the Cl⁻ ions. Moreover, the presence of GGBS in the concrete mix increases the chemical and physical binding of chlorides, contributing to a reduction of the free chlorides in the paste [11].

However, the influence of the amount of GGBS addition seems slightly different for concrete tested at different ages. In fact, the chloride penetration of the specimens tested at 28 days of age decreases gradually as the amount of slag increases. It seems that, at 28 days of age, a significant reduction in the chloride ingress in concrete with GGBS is only achieved for amounts of replacement of about 50% of the Portland cement content. On the other hand, for the specimens tested at 56 days of age, the results show a markedly decrease (47%) in the DRCM coefficient between Portland cement concrete (Mix 1) and concrete with 25% of slag replacement (Mix 2). As the percentage of slag replacement increases, the chloride migration coefficient is not so significantly affected. These results indicate that, for concrete older than 28 days, an increase in the GGBS content does not result in a very significant increase in the chloride resistance.

Globally, these results show that the addition of GGBS is very effective in reducing the chloride penetrability in concrete, when compared to Portland cement concrete, even for replacement levels as low as 25% of the cement content, as long as sufficient hydration of the GGBS has taken place when chloride attack occurs. For earlier ages (at least for 28 days of age), higher percentages of GGBS are necessary to obtain significant reduction in the chloride ingress.

Moreover, even with the slower hydration of GGBS, the results show that the chloride migration coefficient is always lower for slag concrete than for Portland cement concrete of the same age, cured according to the same procedure, even at ages as early as 28 days, and for percentages of replacement as high as 100% of the cement content. These results indicate that, as long as proper curing is provided during the early days, the effect of the addition of GGBS in reducing the chloride ingress in concrete is able to offset the effect of the lower hydration rate and consequent higher diffusivity of the slag concrete at early ages.

### 3.2.2. Influence of the Air Content of Concrete

Figure 14 shows a slight increase in the chloride migration coefficient for Mix 5 (with higher air content), when compared to Mix 3, both at 28 days and at 56 days of age.

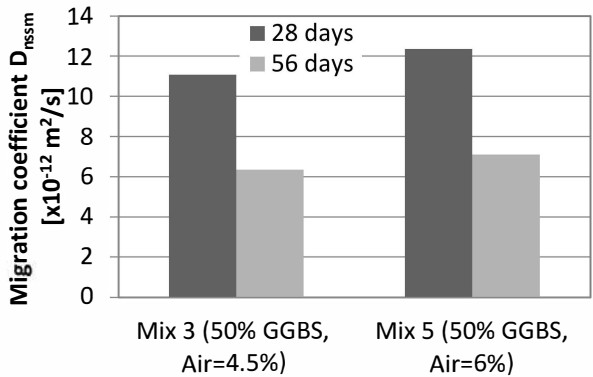

**Figure 14.** Rapid chloride migration coefficient, DRCM, for mixes with 50% GGBS, k = 0.6 and different amount of air, cured at 20 °C.

The chloride ingress in concrete depends mainly on the diffusivity of concrete, which is directly related to its pore structure. It is therefore expected that an increase in the air content will lead to a certain increase in the diffusivity of concrete, due to an increased number of air voids throughout the paste, which may be partially filled with the pore solution. Furthermore, an increase in the number of air pores in concrete due to the use of air entrainment admixtures often results in a reduced spacing factor, which shortens the continuous paths between the air voids in the paste, therefore facilitating the transport of chlorides. On the other hand, the diffusivity of concrete depends not only on the volume and distribution of voids, but also on the size and continuity of the air pores. Given that it is the capillary porosity which controls the diffusivity of concrete (which increases with an increase in the amount and size of capillary pores), and since the air pores entrained using AEA are usually not fully filled with water in the test, the effect of an increase of the pores in the paste on the diffusivity is not as markedly as one would expect. For this reason, only a slight increase in the chloride migration coefficient is obtained for concrete with increased air content (Figure 14).

These results also show that, even though an increase in the air content affects the overall resistance against chloride ingress, it does not influence the development of chloride resistance with ageing of concrete. This fact was expected since, as concluded in the tests of the compressive strength resistance, the air content in concrete affects only its air pore structure, and not the chemical reactions and the rate of hydration of concrete.

### 3.2.3. Influence of the Efficiency Factor

The results presented in Figure 15 show a higher value of the migration coefficient for the concrete specimens with an efficiency factor of 1.0 (Mix 6), both at 28 and at 56 days of age, when compared with Mix 3, with an efficient factor of 0.6. These results were actually expected, considering, as previously referred, that the chloride ingress in concrete is closely related to the diffusivity of the paste, which in turn depends on the water/binder ratio. The equivalent water/cement ratio, determined according to Equation (1) is 0.45 for both mixes. However, for Mix 3 (with an efficiency factor of 0.6), the actual water/binder ratio is 0.39, whereas for Mix 6 (with a k-factor of 1) the actual water/binder ratio is 0.45. Lower water/binder ratios result in lower diffusivity of concrete, which explains why the migration coefficient is lower for Mix 3 when compared to that of Mix 6, although the ratio of GGBS to cement (50%) is the same in both mixes.

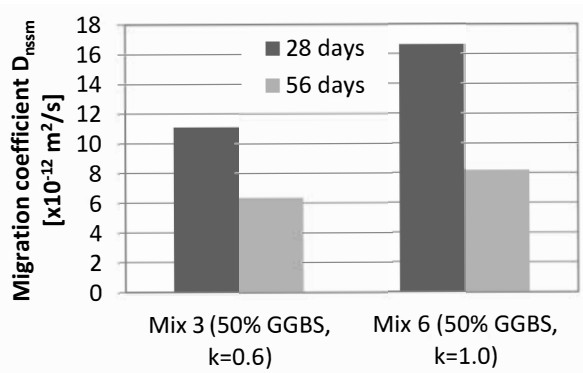

**Figure 15.** Rapid chloride migration coefficient, DRCM, for mixes with 50% GGBS, 4.5% air content and different k-factor, cured at 20 °C.

The increase in the DRCM coefficient for a greater k-factor is more pronounced for concrete tested at 28 days, varying from $11.1 \times 10^{-12}$ m$^2$/s for Mix 3 to $16.6 \times 10^{-12}$ m$^2$/s for Mix 6 (variation of 50%), whereas for concrete tested at 56 days only a slight increase in the DRCM coefficient is identified, increasing from $6.3 \times 10^{-12}$ m$^2$/s for Mix 3 to $8.2 \times 10^{-12}$ m$^2$/s for Mix 6 (variation of 30%). On the other hand, the decrease in the Rapid Chloride Migration coefficient between 28 and 56 days is relatively similar for both concrete qualities (43% reduction for Mix 3 and 51% for Mix 6). It can thus

be concluded that the difference in the k-factor (and consequent difference in the water/binder ratio) does not significantly influence the evolution of the resistance against chloride ingress with the ageing of concrete.

Comparing the results obtained for Mix 6, with 50% of GGBS replacement of the mass of Portland cement (Figure 15) and the results for Portland cement concrete (Mix 1, Figure 13), both with the same actual water/binder ratio (0.45), it can be concluded that, for the age of 28 days, the DRCM coefficient is similar for both concrete qualities. However, the tests performed at 56 days of age reveal a significant lower chloride migration coefficient for concrete with GGBS than for Portland cement concrete. These results are related with the slower hydration reactions of slag, which result in a higher capillary porosity of GGBS concrete at early ages. With continuing hydration, however, concrete with addition of slag presents a much higher resistance against chloride ingress than Portland cement concrete produced with the same w/c ratio.

The results obtained in this investigation show, therefore, that the addition of GGBS in concrete results in a significant improvement of the resistance against chloride ingress in concrete, even for efficiency factors equal to 1, as long as sufficient hydration is ensured.

These results also demonstrate that the efficiency factor concept is an effective way to predict the compressive strength of concrete qualities with additions, but not necessarily its durability. In fact, the parameters required for high-quality concrete are usually related with minimum cement content, minimum cement class, maximum w/c ratio, and compressive strength achieved at 28 days of age. It is usually assumed that concrete that fulfils these requirements will present adequate protection against the aggressive agents. This is partially true. The majority of the deterioration mechanisms depends on the penetration of fluids or gases inside concrete, as it is the case of carbonation, chloride ingress, and frost scaling. Furthermore, as previously explained, the penetration of these agents in concrete depends on its porosity, which in turn depends on the water/cement ratio. There is, therefore, a base for determining an efficiency factor based on an equivalent water/cement ratio. However, this approach does not consider the effects of each addition on the microstructure of concrete–only on the compressive strength development. Therefore, new efficiency factors should be studied considering the durability parameters.

*3.3. Scaling under Freezing and Thawing*

The results from the salt-frost scaling resistance of concrete tested according to the Swedish Standard SS 13 72 44 [9] are presented in this section.

3.3.1. Influence of the Amount of Portland Cement Replacement by GGBS

As shown in Figure 16, the mass of scaled material increased significantly as the amount of GGBS increased. According to the acceptance criteria given by SS 13 72 44 [9], the reference Portland cement-only concrete (Mix 1) showed very good frost resistance after 56 freeze/thaw cycles (presenting a total mass of scaled material below 0.1 kg/m$^2$). The mass of scaled material for Mix 2, with 25% of GGBS was 0.243 kg/m$^2$ after 112 freeze/thaw cycles, which reveals good frost resistance. As for concrete qualities with 50% of GGBS replacement or higher, both Mix 3 (with 50% GGBS) and Mix 4 (100% GGBS) were considered not resistant against salt-frost scaling. Mix 3 failed after 112 freeze/thaw cycles, with an average mass of scaled material of 1.075 kg/m$^2$, whereas Mix 4 (with 100% of GGBS replacement) failed as early as the 56th cycle, reaching a scaling of 1.168 kg/m$^2$. The poor salt-scaling resistance of concrete containing high amounts of GGBS is widely documented, even though some conflicting reports have been obtained over the years, especially between laboratory studies and field investigations. Possible reasons might be the different micropore structures and carbonation resistances between these two types of binder [15].

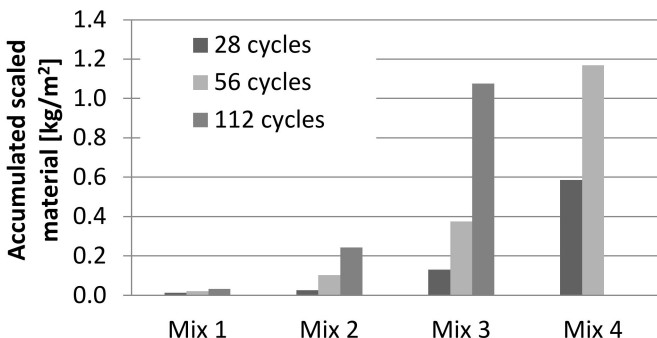

**Figure 16.** Mean values of the accumulated mass of scaled material per area of freezing surface after 28, 56 and 112 cycles for concrete with different amounts of GGBS replacement.

The results obtained support the limitation of 25% of cement replacement by slag prescribed by SS 13 70 03 [5] for concrete exposed to freezing environments where de-icing salts are used. However, it might be possible to increase the percentage of replacement, even for the same air content and efficiency factor. The salt-frost resistance of concrete mixes with replacement levels between 25% and 50% should be investigated. Furthermore, it should be studied if it is possible to produce frost resistant concrete with 50% of GGBS replacement, by making some changes in the mix.

3.3.2. Influence of the Air Content of Concrete

The results plotted in Figure 17 reveal a markedly positive effect of the increase of the air content on the performance of concrete subjected to freeze/thaw cycles in the presence of salts. The total accumulated scaled material for Mix 3 (with targeted air content of 4.5%) after 112 freeze/thaw cycles was 1.075 kg/m$^2$, which is considered not acceptable according to SS 13 72 44 [9]. Mix 6, however, with targeted air content of 6% (and average air content measured according to the pressure method of 5.6%) presented good frost resistance after the end of the test, with an average scaled material of 0.370 kg/m$^2$.

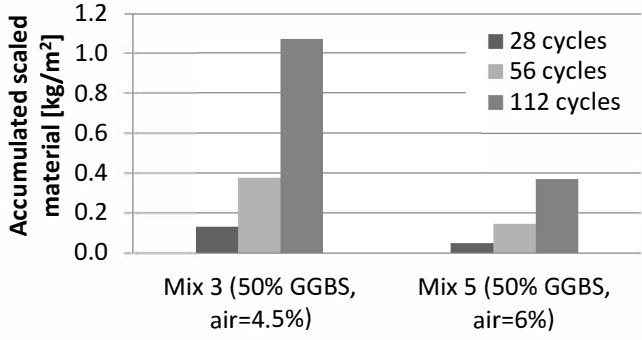

**Figure 17.** Mean values of the accumulated mass of scaled material per area of freezing surface after 28, 56 and 112 cycles for concrete with 50% GGBS and different air content.

According to these results, an increase in the air content from 4.5% to 5.6% changes the salt-scaling resistance of concrete with 50% of GGBS replacement from not acceptable to good. The positive effect of air entrainment in the improvement of the salt-frost resistance of concrete is widely described in the literature. Air entraining agents are used to entrain small-sized, closely-spaced air bubbles in concrete that will accommodate the water expelled from the large capillary pores during freezing, therefore reducing the hydraulic and osmotic pressures caused by freeze/thaw cycles, and, consequently, the chance of damage [12]. In fact, the analysis of the other air pore parameters measured with the AVA shows a clear relation between the increase of air content and the decrease in the spacing factor. The spacing factor changes from a mean value of 0.215 mm for Mix 3 to 0.120 mm for Mix 5. The pore specific surface is also higher for Mix 5 (22.7 mm$^{-1}$) than for Mix 3 (19.8 mm$^{-1}$), which reveals an

average pore size lower for Mix 5 than for Mix 3. It should be noted, however, that the values obtained with the AVA for the total air content of these mixes were markedly higher than those obtained according to SS-EN 12350-7, as presented in 3.1.2. It would be important to determine the actual air content in the hardened concrete mixes, in order to assess if the actual air contents are indeed much higher than the 5.50% and 5.60% values obtained according to the pressure method, or, instead, if it is actually possible to obtain a similar air void structure and adequate salt-frost scaling resistance with an air content between 5% and 6%.

These results show, nevertheless, that it is possible to produce concrete with GGBS replacement levels up to 50% of the cement content that presents good salt-frost resistance, as long as proper air entrainment is provided.

### 3.3.3. Influence of the Efficiency Factor

The results from the salt frost scaling test presented in Figure 18 show that Mix 6, produced with an efficiency factor of 1.0, presents a better performance against salt-frost scaling than Mix 3, with the *k*-factor of 0.6 recommended by SS 13 70 03 [5]. In fact, while specimens of Mix 3 present an accumulated scaled material higher than the acceptable limit of 1.0 kg/m$^2$ at the end of the test, the mean value of the scaled material in the specimens of Mix 6 is under 0.5 kg/m$^2$ after 112 freeze/thaw cycles, which reveals, therefore, good frost resistance.

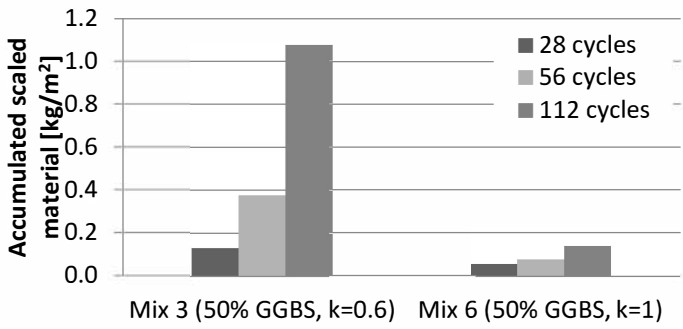

**Figure 18.** Mean values of the accumulated mass of scaled material per area of freezing surface after 28, 56 and 112 cycles for concrete with 50% GGBS and different efficiency factors.

Theoretically, it was expected that Mix 6 would display a lower salt scaling resistance, since an increase in the water/cement ratio usually results in a more porous concrete, thus more able to absorb water, resulting in a higher degree of saturation. Consequently, the hydraulic and/or osmotic pressures inside the paste will increase, resulting in a higher chance of damage due to freeze/thaw cycles [4].

In this case, the pore structure of each of the concrete qualities may have played a more important role in the salt-frost resistance of the mixes than the water/binder ratios and simple porosity of the concrete mixes. The air void parameters shall, therefore, be analysed.

The air content of Mix 6 measured according to SS-EN 12350-7 [8] was the targeted 4.5%. The results of the Air Void Analyser, however, reveal a slightly higher air content for Mix 6 (with 6% for batch number 1) than for Mix 3 (6.2% and 5.5% for batches 1 and 2, respectively). The specific surface measured for Mix 6 was 23.7 mm$^{-1}$, which is higher than the 19.4 and 20.2 mm$^{-1}$ obtained for Mix 3, and closer to the 25 mm$^{-1}$ recommended by the AVA manufacturer. Moreover, the spacing factor obtained for Mix 6 was 0.18 mm, which within the limits recommended by the literature for satisfactory salt-frost resistance of a concrete quality (<0.20 mm), unlike the spacing factor determined for Mix 3, of 0.22 mm.

Considering the results obtained by the AVA for Mixes 3 and 6, it seems that a slight increase in the air content (from 5.85% to 6%) may result in relevant differences regarding the air void structure of concrete, namely an increase in the specific surface and a decrease of the spacing factor. Another possible reason why Mix 6 revealed a better resistance to frost scaling is that Mix 6 (with an effective water-binder

ratio of 0.45) contains more capillary pores with larger sizes than Mix 3 (with an effective water-binder ratio of 0.39). These large capillary pores allow an easy transport of the unfrozen water to the empty air pores. Further investigation should be carried out to determine the capillary suction of the concrete mixes. This would allow assessing whether the worse results of Mix 3, when compared to Mix 6, were actually related to a significant difference in the capillary pore systems of the hardened concrete with GGBS, which may need even smaller spacing factor for adequate transferring the unfrozen water to the air pores. Nevertheless, the results show that it is possible to produce salt-frost resistant concrete with 50% of slag replacement using efficiency factors larger than the recommended by the standards, at least as high as 1.0.

According to these results, it also seems that ensuring an adequate air pore system affects more significantly the performance of concrete under freeze/thaw cycles than the water/binder ratio and the compressive strength of concrete. This fact raises, once again, the question of the adequacy of the use of efficiency factors. In fact, the *k*-factors recommended in the literature and standards were determined solely based on the compressive strength development of a concrete mix with additions, when compared with ordinary Portland cement concrete, by adjusting the water/binder ratio. However, even though the water/binder ratio and the mechanical strength of concrete are important parameters that influence several degradation mechanisms (as observed in the results of this investigation), there may be other parameters that have more influence in the durability of a concrete that are not considered in the efficiency factor concept.

### 3.3.4. Influence of Prolonged Hydration before the Start of the Freeze/Thaw Test

Figure 18 compares the results of the specimens of Mix 3 and Mix 5 tested according to the standard procedure, and hydrated for 14 more days. As previously explained, the test specimens subjected to prolonged pre-treatment were sawn from the unused half of the cubes used in the standard freeze/thaw test of the same concrete qualities. Since the test specimens result from the same cube, the results obtained for the same mix with different pre-treatment procedures are not, therefore, significantly affected by mechanical properties and different air pore structures.

As seen in Figure 19, the specimens subjected to prolonged hydration before the freeze/thaw test generally present a lower mass of scaled material than the specimens tested at 31 days of age, for both Mixes 3 and 5. The specimens of Mix 3 subjected to prolonged pre-treatment show a better performance than the specimens pre-treated according to the standard at all ages. In fact, the specimens of Mix 3 pre-treated according to the standard fail after 112 cycles, whereas the mix that was allowed to hydrate longer presents acceptable frost resistance at the end of testing.

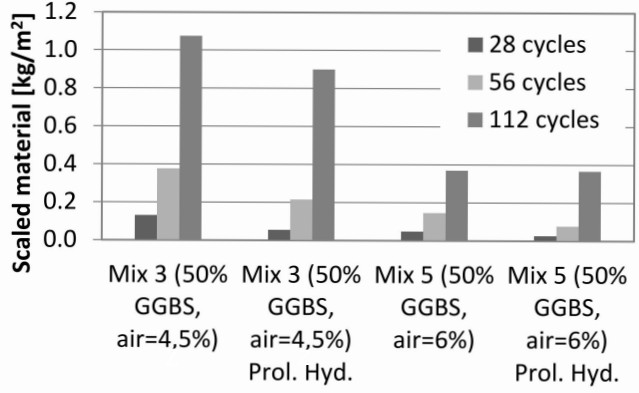

**Figure 19.** Comparison between the scaled material of Mixes 3 and 5, pre-conditioned according to the standard procedure and hydrated in the climate chamber during more 14 days.



As for Mix 5, the specimens tested at 45 days of age reveal a slight better frost resistance throughout the test, but not as evident as for Mix 3. However, at the end of the test, the accumulated scaled material is approximately the same for both cases.

These results were somewhat expected. On the one hand, the specimens that remained 14 more days in the climate chamber were allowed to cure (moist curing at 20 °C, 65% R.H.) for a longer period before the first freeze/thaw cycle. This means that the hydration degree of these specimens was higher at the start of the test, i.e., the specimens presented higher compressive (and tensile) strength, and reduced amount of capillary porosity, which positively influences the salt-frost resistance of concrete. On the other hand, the specimens subjected to prolonged pre-treatment in the climate chamber were not in contact with water, which may have allowed prolonged drying of the test specimens before the first freezing took place (21 days, instead of 7 days for the specimens pre-conditioned according to SS 13 72 44). In fact, according to Neville [4], drying of concrete before the exposure to freeze/thaw cycles improves its scaling resistance, as long as adequate wet curing was provided prior to drying, in order to ensure extensive hydration. Drying of the specimens lowers the degree of saturation of concrete, which means that the critical degree of saturation is not reached as early as for mixes which were not allowed to dry, and, therefore, less damage is obtained.

This may, in turn, explain why the mass of scaled material of the specimens tested at 45 days of age is significantly lower than the specimens tested at the $31^{st}$ day of age in the first freeze/thaw cycles, but not as much in the last cycles. As seen in Figure 18, the mean accumulated scaled material of the concrete qualities subjected to prolonged pre-treatment increases significantly after 56 cycles, with Mix 5 with prolonged hydration reaching the same value of Mix 5 without prolonged hydration after 112 cycles. This may be related to the lower degree of hydration of the mixes tested at 31 days of age at the start of the test, which critical degree of saturation is achieved earlier, resulting in increased damage in the early cycles. After the critical degree of saturation is reached also for mixes tested at 45 days of age, the effect of the freeze/thaw cycles in the presence of salts is similar for mixes tested at both ages.

The positive influence of the prolonged pre-treatment in the climate chamber seems more pronounced for Mix 3 than for Mix 5. This fact may be related with the different air content of the two mixes. The targeted air content for Mix 3 was 4.5% ± 0.5% (with an actual average air content between the two batches of 4.5% according to SS-EN 12350-7), and 6% ± 0.5% (with an actual average air content of 5.6%) for Mix 5. The results show, therefore, that the influence of the degree of hydration of concrete in the protection against frost scaling is more significant for concrete with lower air contents. This would be expected, since concrete with lower air contents have less voids to which the water can move in when ice starts to form. The critical degree of saturation is, therefore, reached earlier. Research carried out by Utgennant [15] showed that drying has a relatively small, but usually positive effect on the scaling resistance of both Portland cement concrete and GGBS concrete, especially at early ages (up to 31 days of age). The author also found that this positive effect is more pronounced for GGBS concrete than for Portland cement concrete, and that it increases with an increase in slag content.

These results show that, after adequate curing, GGBS concrete structures shall be allowed to dry for a long period of time before being exposed to the first freeze/thaw cycle.

These results also raise a question of the applicability of the salt-frost scaling test described in SS 13 72 44 [9] for concrete with additions. This test method was developed for Portland cement concrete, and the first freeze/thaw cycle occurs at 31 days of age. The maturity of Portland cement concrete does not increase significantly after 28 days of age [4], which means that the age of testing is probably adequate for Portland cement concrete. However, for concrete with additions, the maturity of concrete continues to develop long after the 28 days of age, depending on the type, reactivity and amount of addition. Therefore, there is a possibility that this test method underestimates the salt frost resistance of concrete qualities with additions. One possibility could be to start the test at a later age. This subject should be further investigated.

It should also be pointed out that carbonation has a negative effect on the frost resistance of concrete with additions of GGBS exceeding approx. 50% of the cement weight, see e.g., [16–19]. In all these studies it has been shown that carbonation reduces the frost resistance of the concrete mainly due to the pore system of the concrete gets coarser from carbonation. Therefore it has been suggested to revise the pre-conditioning of the concrete samples, where concrete containing more than 25% GGBS shall be subjected to pre-carbonation before testing according to SS 13 72 44 [9]. Example of such a procedure is described in [17] and [18].

## 4. Conclusions

From the results of this investigation, a general conclusion can be drawn: concrete with addition of GGBS (fulfilling SS-SS-EN 15167-1 [6]) up to 25% of replacement of cement (maximum amount of replacement permitted by SS 13 70 03 for exposure class XF4) presents adequate salt-frost resistance.
Other conclusions are summarized below:

- The frost resistance of concrete generally decreases with an increase of the addition of GGBS, for concrete with 4% to 5% of air content by volume. This fact may be due to the slower hydration of GGBS when compared to Portland cement concrete, which yields a more porous concrete with a lower strength at the age of the start of the freeze/thaw test. The results showed, however, that it is possible to produce frost resistant concrete with GGBS amounts up to 50% (of the weight of CEM I), by changing some properties of the mix (such as increasing the air content), i.e., it is possible to produce salt-frost resistant concrete with percentages of GGBS replacement higher than the limit defined by SS 13 70 03 [9] for exposure class XF4 (25% of the weight of CEM I).
- For concrete with 50% of the weight of cement replaced by GGBS, the results for frost resistance after 112 cycles changed from not acceptable, for concrete with an air content of 4.5%, to good, for concrete with 5.6% of air. These results show that the beneficial effect of an adequate air pore structure in the salt-frost resistance of concrete is also valid for concrete with GGBS. The results also showed that amounts of GGBS up to 50% of the cement weight can be safely used in freezing environments where de-icing salts are used, as long as proper air entrainment is provided.
- Addition of GGBS in concrete significantly improves the resistance against chloride ingress, when compared to Portland cement concrete. The results show that the performance of concrete against chloride penetration increases with increased cement replacement levels (up to 100% of the Portland cement weight). The resistance of GGBS concrete against chloride is mainly related to its denser and more refined microstructure, which results in a less permeable concrete and, consequently, on a slower diffusion of the Cl$^-$ ions. On the other hand, GGBS has also shown to improve both the physical and chemical binding of chloride ions, which also contributes to a reduction of the free chlorides in the concrete paste.
- The results also show that GGBS concrete outperforms Portland cement concrete at all ages, and all percentages of replacement, even if the slag replaces the cement on a one-to-one basis (i.e., for a *k*-factor of 1.0). The efficiency factor concept is based on the water/binder ratio and compressive strength development of a concrete quality with additions, in comparison to Portland cement concrete, while no durability aspects are taken into account. The present results raise questions about the applicability of the efficiency factors when durability issues under concern.

**Author Contributions:** Conceptualization, J.G.F., L.T. and A.L.; methodology, V.C., L.T. and A.L.; validation, V.C. and L.T.; formal analysis, V.C.; investigation, V.C.; resources, L.T. and A.L.; data curation, V.C.; writing—original draft preparation, V.C.; writing—review and editing, J.G.F., L.T.; visualization, V.C.; supervision, J.G.F., L.T.; project administration, A.L.; funding acquisition, A.L. All authors have read and agreed to the published version of the manuscript.

**Funding:** This research received no external funding.

**Acknowledgments:** The authors grateful acknowledge C-Lab Thomas Concrete Group for the support given to the experimental campaign developed in this research.



**Conflicts of Interest:** The authors declare no conflict of interest.

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
