# Peer review of "Effect of the Addition of GGBS on the Frost Scaling and Chloride Migration Resistance of Concrete"

_applsci, doi:10.3390/app10113940_

Round 1

Reviewer 1 Report

The introduction is concise and justifies the need for research. However, there is a lack of references for previous work in the area and the limited number of references were used to back-up the information provided in the manuscript.

The mix design (proportions) presented in section 2 is not clear. A table containing all mixes would be recommended. Please clarify the amount of cement, GGBS, aggregates, admixture, and water for each mixture.

Authors mentioned that ‘Air entrained concrete mixes were 66 produced and tested with different amounts of replacement, namely 0%, 25%, 50% and 100% of GGBS 67 by weight of CEM I.’ also that ‘CEM I 42.5 N MH/SR/LA’was used in this research. After that, authors present the following explanation:

‘Mix 2 - 25% GGBS (which is the maximum amount of GGBS allowed for XF4 exposure class  according to SS 13 70 03 [4]). This corresponds to an addition of 20% GGBS of the binder weight (i.e. a CEM II/A-S);

Mix 3 - 50% GGBS. This corresponds to an addition of 35% GGBS of the binder weight (i.e. a  CEM II/B-S);

Mix 4 - 100% GGBS. This corresponds to an addition of 50% GGBS of the binder weight (i.e. a  CEM III/A).

The information provided is confusing and does not explain how the mixes were produced and the amount of replacement used.

Clarify number of samples tested for each of the tests (line 134 to 136)

Rapid chloride migration test, according to NT Build 492; and  Salt-frost scaling, according to SS 13 72 44 [7] are not typical tests. More information about both testing procedures and samples used are required.  Add also sample dimensions.

The authors mentioned that ‘One slab was pre-conditioned 146 according to the standard procedure described in SS 13 72 44. The other slab was kept in the climate 147 chamber for 14 more days, and only afterwards the water was poured onto the test surface. For these 148 specimens, the test started at the 45th day of age, instead of the 31st day.” It is unclear which pre-conditioning procedure was used.

Authors mentioned that ‘The results show that the concrete quality with higher amount of air (Mix 5, with air content 195 between 5.5% and 5.6%, measured according to SS-EN 12350-7) presents lower compressive strength’. What do authors mean by ‘concrete quality;’?

In line 253 authors mentioned ‘an efficiency factor concept was developed.’ In line 99 authors explained that ‘the mass of slag replaces the exact same reduced mass of Portland cement.’, but the information is very confusing. Please clarify the meaning of efficiency factor and the differences between the two values used in the manuscript: 0.6 and 1.

Line 291: Authors mentioned that ‘The positive influence of the increase in GGBS content in the improvement of the resistance against chloride migration is widely reported, and is usually attributed 293 to the following factors’. There are several hundreds of published papers on this field, and at least the most important ones should be acknowledged here, instead of only using Neville as reference.

The explanations presented in section 3.2 and 3.3 about the testing procedure should not be in the results section, instead they should be in section 2, as they are methods of testing and not results.

Authors must clarify methods/test procedure, mix design (proportions), number of samples, cure conditions, and types of exposure to allow a proper review of results and conclusion.

Reviewer 2 Report

Authors need to rectify the following comments before accepting the manuscript:

  1. In the introduction part, authors mentioned the influence of  freezing environment on concrete in the presence of de-icing salts. However, two of the most important references that discuss this issue thoroughly are missing. Please, include the following references: (1) Al-Kheetan, M.J. and Rahman, M.M., 2019. Integration of anhydrous sodium acetate (ASAc) into concrete pavement for protection against harmful impact of deicing salt. JOM, 71(12), pp.4899-4909. (2) Al-Kheetan, M.J., Rahman, M.M., Ghaffar, S.H. and Jweihan, Y.S., 2020. Comprehensive Investigation of the Long-term Performance of Internally Integrated Concrete Pavement with Sodium Acetate. Results in Engineering, p.100110. 
  2. In the Laboratory study section, authors mentioned that low-alkali cement was used. Please indicate the reason behind using a low-alkali cement? 
  3. In the laboratory study section, authors controlled the workability of the mix by adding superplasticizer. Adding superplasticizer would affect the compressive strength of the tested mixes which will be reflected on the comparison between them (superplasticizer will increase compressive strength at 28 days and 56 days by enhancing the effectiveness of compaction to produce denser concrete). This was clear in section 3.1.1. Please, discuss this issue in section 3.1.1 
  4. Figure 5 is hard to read. Please make it clearer
  5. In section 3.1.1, please explain the interaction mechanism between GGBS and the other materials in the mixes that led to similar 28-day compressive strength for all mixes. 
  6. In section 3.2.1, please discuss the effect of adding superplasticizer on the chloride migration results. 
  7. In section 3.3.1, what is the reason behind increasing the scaled material with increasing the addition of GGBS ? 

Reviewer 3 Report

The topic of this document is very interesting and the experimentation is well done. However, some minor and major revisions are needed to improve the quality of the paper.

Line 104: “Summarizing, a total of eight different mixes were produced”. The authors listed only six mixes under this sentence and the mixes actually appear to be six.

Figures 5-7. The scale used for the horizontal axis is unclear. Please explain which scale was used.

Line 222: “as shown by the similar slope of the lines of Figure 6 for both concrete qualities”. This claim is inadequate. If the slopes were similar, an increase in the air content of concrete would affect the rate of strength development, since the strength values are different. In order to show that the rate of strength development is almost the same for the two air contents, the authors should extend the segments linking the data collected after 28 and 56 days and show that these two lines intersect at a point that is located on (or near to) the horizontal axis. The same could be done for the pair of values after 7 and 28 days, even if the intersection point is probably more distant from the horizontal axis, this second time. Please change the sentence in line 222 and add one or two figures to discuss the rate of strength development more correctly.

Line 252: “the rate of strength gain is comparable for both mixes”. Add one or two figures to support this claim, as described above.

Line 356: “Lower water/binder ratios result in lower diffusivity of concrete, which explains why the migration coefficient is lower for Mix 3 when compared to that of Mix 6”. In the reviewer’s opinion, even the larger amount of addition in Mix 3 could have contributed to the reduction of the migration factor, comparing Mix 3 with Mix 6. Please add a discussion on this.

Line 417: “According to the acceptance criteria given by the SS 13 72 44 [7]”. Could the authors specify what the acceptance threshold is, according to SS 13 72 44?

Line 448: “According to these results, an increase in the air content from 4.5% to 5.6% changes the salt-scaling resistance of concrete with 50% of GGBS replacement from not acceptable to good”. This is difficult to say in the absence of the reference value provided by the SS 13 72 44. Please specify the acceptance threshold.

Line 458: “It should be noted, however, that the values obtained with the AVA for the total air content of these mixes were higher than those obtained according to SS-EN 12350-7”. How much greater? Please explain.

Reviewer 4 Report

Main text and number of figures might be a bit too much, but other than that this is a well written manuscript and I recommend publication of this article in this journal. 

Round 2

Reviewer 1 Report

Dear authors,

The manuscript can be accepted in the current format

Reviewer 3 Report

The authors edited the text consistently with the reviewer's comments. This reviewer has no further requests.